# An Innovative Polymeric Platform for Controlled and Localized Drug Delivery

**DOI:** 10.3390/pharmaceutics15071795

**Published:** 2023-06-23

**Authors:** Monica Elbjorn, Jacob Provencio, Paige Phillips, Javier Sainz, Noah Harrison, David Di Rocco, Ada Jaramillo, Priya Jain, Alejandro Lozano, R. Lyle Hood

**Affiliations:** 1Department of Mechanical Engineering, University of Texas at San Antonio, San Antonio, TX 78249, USA; monica.elbjorn@my.utsa.edu (M.E.); jacob.provencio@my.utsa.edu (J.P.); javier.sainz@my.utsa.edu (J.S.); noah.harrison@my.utsa.edu (N.H.); david.dirocco@my.utsa.edu (D.D.R.); priya.jain@my.utsa.edu (P.J.); 2Department of Biology, University of Texas at San Antonio, San Antonio, TX 78249, USA; paige.phillips@my.utsa.edu; 3Department of Biomedical Engineering, University of Texas at San Antonio, San Antonio, TX 78249, USA; ada.jaramillo@my.utsa.edu; 4Tecan, Morrisville, NC 27560, USA; 5Department of Obstetrics & Gynecology, The University of Texas Health Science Center at San Antonio, San Antonio, TX 78229, USA; lozanoa3@uthscsa.edu

**Keywords:** polymeric drug delivery, targeted drug delivery system, polycaprolactone, poly-lactic acid, rhodamine B, polymer casting, cervical cancer

## Abstract

Precision medicine aims to optimize pharmacological treatments by considering patients’ genetic, phenotypic, and environmental factors, enabling dosages personalized to the individual. To address challenges associated with oral and injectable administration approaches, implantable drug delivery systems have been developed. These systems overcome issues like patient adherence, bioavailability, and first-pass metabolism. Utilizing new combinations of biodegradable polymers, the proposed solution, a Polymeric Controlled Release System (PCRS), allows minimally invasive placement and controlled drug administration over several weeks. This study’s objective was to show that the PCRS exhibits a linear biphasic controlled release profile, which would indicate potential as an effective treatment vehicle for cervical malignancies. An injection mold technique was developed for batch manufacturing of devices, and in vitro experiments demonstrated that the device’s geometry and surface area could be varied to achieve various drug release profiles. This study’s results motivate additional development of the PCRS to treat cervical cancer, as well as other malignancies, such as lung, testicular, and ovarian cancers.

## 1. Introduction

International sources reported 604,127 newly diagnosed women with cervical cancer and 341,831 deaths related to cervical cancer in 2020 [1,2]. Of the deaths reported, 90% occurred in low- and middle-income countries [3]. Although there have been significant technological developments in the last 40 years, including Pap smear and human papillomavirus (HPV) screening, as well as HPV vaccines, the mortality rate for patients with cervical cancer remains at approximately 30% [3,4]. In the United States, the American Cancer Society projects that in 2023, approximately 13,960 new cases of invasive cervical cancer will be diagnosed. Patients diagnosed with early-stage tumors can be successfully treated with a conservative surgical approach compared to patients with more advanced cervical cancer, who are instead treated with radiotherapy and concurrent chemotherapy [4,5]. Clinical treatment of recurrent metastatic cervical cancer can vary based on the patient’s medical history, genetics, the location of the malignancy, and the time of recurrence, presenting a challenge for clinicians [3,4,5,6]. Personalized medicine approaches tailored to an individual patient’s genetic and phenotypic nuances are a shared vision across medicine. Although cervical cancer treatment is selected on a case-specific basis, new sustained drug delivery systems for treating a localized area are urgently needed to increase the survival rate for metastatic and/or recurrent cervical cancer in patients [7,8,9]. 

For years, radiation therapy has been the primary localized treatment option for clinicians to treat advanced cervical cancer. The ability to combine chemotherapy and radiotherapy to treat localized cervical cancer depends on the dosage required to treat a large tumor, without exceeding concentrations associated with toxicity [10,11]. Recent randomized controlled trials have demonstrated the effectiveness of adding systemic cisplatin and radiation for localized cervical cancer. The results indicated improved rates of survival among patients with the localized treatment relative to radiation alone [11,12,13]. Cisplatin is an antineoplastic agent that prevents the growth of cells and is often used to treat solid tumors [14,15,16]. The drug works by interfering with natural repair mechanisms to activate signal transduction pathways and cause cell death [17,18,19]. Despite its wide range of applications as an anticancer agent, cisplatin can cause side effects, such as impaired hearing, difficulty breathing, pain, change in frequency of urination, drowsiness, and unsteadiness. These side effects motivate the local delivery of cisplatin to minimize adverse events.

Targeted medicine aims to enhance conventional pharmacological treatments by considering a patient’s genetic and environmental factors, thereby enabling the customization of medication dosages [17]. However, the oral delivery of drugs presents notable challenges, including issues related to first-pass metabolism, suboptimal properties, and adverse effects, which can adversely impact patient compliance. Implantable drug delivery devices have emerged as a promising alternative system for the treatment of cervical cancer, offering numerous advantages over traditional drug delivery methods.

For instance, Yu Han et al. reported that cisplatin stands out as an exceptionally effective chemotherapeutic agent globally. Nonetheless, the clinical utilization of cisplatin faces significant obstacles in the form of resistance and toxicity, leading to limitations in its application. Despite remarkable progress in precision medicine and immunotherapy, cisplatin-based treatment protocols remain the foremost choice for addressing diverse solid tumors, including those impacting the head and neck, lungs, testicles, ovaries, and bladder [20]. The available evidence indicates that effective therapeutic outcomes require targeted cisplatin release for the treatment of cervical cancer. Dasari et al. prepared a polyethylene glycol (PEG) implant with a mass ratio of 80:20 PEG3350 and PEG400. The drug delivery vehicle was implanted into the cervical cavity of a murine tumor model, and released cisplatin locally. The implant was designed to provide a fast release following implantation to act as an adjuvant to radiotherapy. The study concluded that the localized implant provides a biodistribution profile with superior tumor drug accumulation than a systemic circulation carboxyphenoxy propane-co-sebacic anhydride (CPP-SA) implant to deliver cisplatin intratumorally. The loaded implant was placed into a murine fibrosarcomata model, and again demonstrated a better response than systemic treatment [13,18]. An additional example is the work of Rajkumar et al. and the human ovarian carcinoma SKOV-3 xenograft model. The polyethylene glycol-poly-(l-glutamic acid) dithiodipropionic-Pt (PEG-P(LG-DTDP-Pt) micelles loaded with cisplatin [21,22,23]. These studies have demonstrated that localized methods for delivering cisplatin have the potential to improve therapeutic efficacy by increasing drug accumulation in the affected area while reducing adverse side effects [24,25,26,27]. However, despite the advancements in these studies, none have been able to effectively demonstrate other crucial components in localized drug delivery. Current administration of cisplatin locally lacks a biodegradable controlled release system utilizing novel geometries to enable versatile modulation of dosage.

Motivated by the need for improved cervical cancer treatments and the advancements made in the localized delivery of cisplatin, this study explores a new polymeric drug delivery platform for controlled delivery. Termed the Polymeric Controlled Release (PCRS), the hypothesis is that the platform can provide controlled administration over a multi-week period through designed geometry and surface area. The objectives of this first study on the technology were to demonstrate that these devices could be repeatably fabricated and that they provide a biphasic linear controlled release profile in vitro, which together would indicate the platform’s potential to serve as an effective treatment vehicle for cervical malignancies. Rhodamine B (RhB) was chosen as an analog of the hydrophilic drug cisplatin in this study, as it has a similar hydrodynamic radii, is highly water soluble, and is easily detected through spectroscopy. A polymer injection mold technique was developed and validated for manufacturing the PCRS devices. Experiments examining the RhB release of different PCRS geometries in a phosphate-buffered saline (PBS) solution were conducted to assess release rates and profiles.

## 2. Materials and Methods

### 2.1. Biomaterials

The two biocompatible polymers used in this study were polycaprolactone (PCL) and polylactic acid (PLA), which were mixed with RhB to form the PCRS devices. The PCL was approved by the Food and Drug Administration (FDA) to be utilized as an implantable device due to its high biocompatibility and harmless biodegradation [28,29,30]. It was selected due to its melting point of 60 °C, malleability, and capacity to be 3D printed [31,32,33]. The PCL in pellet form with a molecular weight of 80,000 Mw (114.14 g/mol) was purchased from Sigma-Aldrich^®^. The PLA is a biodegradable polymer that has exhibited thermoplastic behavior and strong tensile properties. This FDA-approved polymer has demonstrated utility in applications such as drug carriers, medical tools, and bone engineering [34,35]. Due to its hydrophilic properties and the operational melting point of 180 °C, PLA was investigated as a PCRS base in this study. The PLA in pellet form with a molecular weight of 60,000 Mw (230 g/mol) was purchased from Filabot^®^. Rhodamine B (RhB) is a tracer dye that was selected to act as a drug analog for cisplatin in this study. RhB has a melting point of approximately 211 °C and a molecular weight of 479.01 g/mol [36,37,38]. UV–Vis spectroscopy was used to determine RhB’s absorbance in an aqueous solution at a wavelength of 553 nm [36]. RhB was purchased from ARCOS Organics (code 132311000). Phosphate-buffered saline (PBS) is a buffer solution commonly used as an analog for interstitial fluids, as it is aqueous, isotonic, and has a pH of 7.4 [23,39]. PBS in powder form (9.940 +/− 0.153 g) was mixed with 1000 mL of de-ionized water from Millipore water purification system (ELGA^®^/PURELAB) to create the buffer solution [40]. The PBS powder was purchased from Avantor^®^ (PBS Buffer Powder PK100).

### 2.2. Polymer Casting Molding

Figure 1 shows the injection mold techniques used to manufacture the devices being tested. The goal of the polymer casting process was to achieve reproducible devices with tight tolerances [41,42,43]. A double-sided aluminum mold was fabricated with a feeder channel (diameter = 6.350 mm, depth = 12.700 mm). This channel led to interlocking channels connected to twelve sphere cavities (2.032 mm) and one rod cavity (8.255 mm × 2.032 mm). The mold was designed to manufacture a small batch of PCRS spheres and an PCRS rod simultaneously (Figure 1A). The production process was initiated by weighing out 1.980 g (+/−0.0001 g) of the polymer (PCL or PLA) being tested and 0.020 +/− 0.0001 g of RhB with a ratio of 99 parts of polymer and 1 part of RhB. Next, the polymer was mixed with RhB by elevating the temperature of polymers to their crystallization point (Figure 1B). The aluminum mold was clamped together to ensure a tight fit, then heated to 80 °C with a heat gun. After holding the mold at that temperature for 3 min, malleable polymer was fed into the entry channel and tamped into the mold with a 6.35 mm metal rod until it extruded into the atmosphere through the side runner. For one hour, the mold was cooled at room temperature (Figure 1C). Once cooled, the polymer casting devices were removed (Figure 1D). The devices were processed, visually inspected for any defects, and weighed for validation (Figure 1E). The PCRS sphere device is shown in comparison against a US dime (Figure 1F). To evaluate the surface quality of fabricated devices, visual inspection using a microscope (LW Scientific/10X/22) was employed for evaluation of surface defects, such as air bubbles, excess material, texture, underfill, and any casting uniformity. The devices were measured with calipers, and we only measured devices at dimensional tolerances under +/−0.020 mm to assure geometrical consistency and device uniformity for testing.

### 2.3. Controlled Release Experimental Protocol

Following the batch manufacture through an injection mold process, RhB-loaded polymer devices (PCL and PLA) were tested for their release rate of RhB in PBS in vitro. Devices were submerged in 150 mL of PBS that was maintained at 37 °C in an incubator (VWR^®^/Gravity Convection Incubators). The PBS sink solution was continuously mixed via a magnetic stir bar at 250 rpm. Samples of 2.5 mL sink solution were removed with replacement at the given time interval for the experiment (hourly or daily). The 2.5 mL samples were analyzed through UV–Vis spectroscopy (Perkin Elmer^®^/LAMBDA 365r) at a wavelength of 553 nm [36,44]. Several different experiments were conducted following this common protocol or with minor modifications based on experimental needs as described below.

### 2.4. Mixture Homogeneity Tests

An experiment was conducted to ascertain the homogeneity of PCRS sphere devices by testing the polymer/RhB mixture of the batch-manufactured device. A 2.032 mm PCRS sphere was manufactured, weighing approximately 0.0048 g, and was divided into four sections. The PCRS sections (n = 4) were submerged into 150 mL of PBS for 12 days. Daily samples of 2.5 mL were taken with replacement for UV–Vis spectroscopic quantification. Samples were collected daily, handled, and analyzed as previously described. As a secondary experimental validation, quartered devices without submersion were imaged using focused ion beam-scanning electron microscopy (FIB-SEM) at a series of magnifications to provide both a qualitative and quantitative assessment of homogeneity. For SEM inspection, PCL PCRS spheres were split in half with a scalpel, then treated with a carbon sputter coater (SPI/Carbon Coater) to prevent the charging effects of non-conducting surfaces. Samples were processed to analyze the inner microstructure through a FIB-SEM under various magnifications 33×, 762×, 2010×, and 4650×.

### 2.5. Initial Release Experiments

An hourly experiment was conducted to capture the initial response of devices submerged in PBS with higher granularity on the first day following submersion. The procedure involved submerging a PCRS sphere (PCL or PLA) in 150 mL of PBS with hourly sampling for 12 h. Samples were collected hourly, handled, and analyzed as previously described.

### 2.6. Single Spherical Device Experiments

The release rate of RhB was assessed in vitro from spheres made of PCL or PLA in PBS. These five PCRS spheres had a diameter of 2.032 +/− 0.02 mm and weighed 0.0060 +/− 0.0001 g. Trials were conducted for 21 days (n = 5) for both PCL and PLA. Samples were collected daily, handled, and analyzed as previously described.

### 2.7. Geometry Variance Experiments

Experiments were conducted to explore the impact of varying implant geometry on the release rate. Four geometries were selected and are exhibited in Figure 2. The geometries compared were a PCRS sphere with a 2.032 mm diameter (Figure 2A); a 2.032 mm sphere with 0.635 mm through-hole (Figure 2B); a 5.080 mm length and 2.032 mm diameter cylindrical rod (Figure 2C); and a 5.080 mm length and 2.032 mm diameter cylindrical tube with a 0.635 mm bore (Figure 2D). Samples were collected daily, handled, and analyzed as previously described.

## 3. Results and Discussion

### 3.1. Casting Polymer Reproducibility Analysis

Each device was weighed after fabrication, and only those that were within +/−100 μg of the target for the various geometries, as described in Table 1, were used in the experiment. Those devices outside these allowed tolerances were discarded.

Figure 3 shows the FIB-SEM visualizations of RhB particles (bright white) that were mixed with PCL. An analysis was performed to process the FIB-SEM images to quantify the amount of RhB per surface area using ImageJ. The images were enhanced with a contrast and converted to 8-bit for calculation, then binary thresholding parameters were used to prepare the image for inversion through adjusting the threshold at 33.30% to emphasize the particles. Evaluation proceeded by converting the number of white pixels and the number of particles inside the area. The results displayed approximately 25% in the selected surface area of the FIB-SEM images with RhB particles, indicating a thorough and homogenous mixture between PCL and RhB.

### 3.2. Mixture Homogeneity Release Analysis

Table 2 shows the experimental results obtained from the mixture homogeneity release study. For this, a PCRS spherical device was split into four sections weighing approximately 1.500 +/− 0.20 μg. Analysis of the daily samples taken over the 12-day period demonstrated the release followed a biphasic linear profile, as seen in Figure 4. The mass release plot of RhB exhibits a larger initial burst of RhB on the first day, followed by an incremental increase for the remainder of the experiment. The initial burst is apparent through the first day of sample collection, with an RhB release of 0.596 +/− 0.11 μg for the first day. During the second phase (day 2–12) of the experiment, the RhB demonstrated an incremental linear constant release of 0.007 μg/mL over the subsequent 11-day period, showing an R^2^ = 0.9323; the tested sections reached a final RhB release of 1.523 +/− 0.08 μg over the entire 12-day period.

Figure 4 displays the mixture homogeneity data for the single sphere divided into four parts used in this experiment. Through the similar release of the pieces, it can be concluded that the polymer and RhB composition is homogenously distributed throughout each device.

### 3.3. Initial Release Analysis

An experiment examining the first 12 h of release at hourly intervals was conducted on five spherical PCRS testing devices (n = 5). Figure 5 exhibits very little release in the first hour (RhB release of 0.097 +/− 0.02 μg), with a jump in release rate at the second hour (cumulative RhB release of 0.523 +/− 0.03 μg) before settling to a steady rate from hours 3–12 (cumulative RhB release of 1.359 +/− 0.08 μg). A linear regression of the first 3 h was associated with an R^2^ = 0.9695. A linear regression of hours 3–12 was associated with an R^2^ = 0.9945.

With the data exhibiting a linear loading phase in the release of RhB, further examination of a polymeric base delivery system should be conducted. Further examination of a polymeric base delivery system is recommended to determine its potential as a viable option for systemic protocols in the treatment of cancer patients. This study highlights the importance of understanding the release kinetics of drug delivery systems, particularly during the initial phases of release [45,46].

### 3.4. Single Spherical Device Analysis

The cumulative RhB release of PCRS spherical devices, both PCL-based and PLA-based, were tested for 21 days (n = 4), as shown in Figure 6. The RhB release profile from PCL can be divided into two linear phases: an initial burst release, and a later sustained release. Examining the spectroscopy data, the initial burst from the PCL devices lasted one day. The cumulative RhB release at the end of day 1 was 2.295 +/− 0.40 μg, equivalent to 0.04% of the total RhB loaded into a singular device. After the initial phase release, the secondary release phase continued at an approximate constant rate of 0.11 μg/mL. The total cumulative RhB release in the second phase was 4.569 +/− 0.30 μg, equivalent to 0.08% of total RhB release from the device. A linear regression of days 1–21 provided an R^2^ = 0.9963. Immediately following immersion of the device into the PBS, the polymer begins to swell, increasing the internal pore size, and increasing the rate of RhB release [33,41,42]. However, PLA demonstrated no release at all in vitro, which was attributed to its poor solubility and hydrophobic side chain [47]. The hydrophobic side chain (methyl groups) causes PLA to have lower water absorption properties and slower degradation. However, the PLA spherical devices failed to release RhB when implanted in vitro. For this study, 21 days were chosen as the release period, which may have been insufficient for the PLA devices. In other studies, PLA has demonstrated complete degradation and total loss of mass over 24–30 months [45,46]. Though PLA did not exhibit any release over the three-week period, these results raise the possibility of creating a PCL–PLA composite structure that can be leveraged to control aqueous access to PCL mixed with a drug for dosage and timeframe control.

### 3.5. Geometry Variance Analysis

Experiments investigating the variation in release rate for the four PCRS geometries (sphere, sphere with a through-hole, cylindrical rod, and cylindrical tube) suggested that RhB release was dependent on device surface area. The data is shown in Figure 7, wherein each of the four PCRS geometrical devices were tested with five replicates for 21 days. An increase in the surface area of the device resulted in a corresponding increase in the amount of RhB released. The overall surface area of each device was calculated from initial measurements after fabrication. Changes in surface area following submersion during the three-week period were not considered, as removing the devices from solution for measurement would impact the release dynamics.

In summary, the data indicated that the PCRS cylindrical tube, with a surface area of 48.387 mm^2^, reached an overall RhB release of 42.068 +/− 0.389 μg for RhB release over the 21-day period. Second was the PCRS cylindrical rod with a surface area of 38.903 mm^2^, reaching an RhB release of 27.129 +/− 0.220 μg of RhB at the end of day 21. The PCRS sphere with a through-hole, with a surface area of 34.838 mm^2^, had an overall RhB average release of 12.392 +/− 0.123 μg at the end of the experimentation period. The solid sphere, with a surface area of 12.968 mm^2^, demonstrated the lowest final RhB release of 5.079 +/− 0.316 μg for day 21. All geometries demonstrated a similar biphasic quasi-linear release, as had previously been seen for the spherical devices, and were segregated from days 0–1 and days 1–21. The PCRS sphere, the sphere with the through-hole, the rod, and the tube exhibited an R^2^ = 0.9742, 0.9782, 0.9972, and 0.9917, respectively. A one-way ANOVA test was performed to evaluate statistical significance between the mass release rates from the different geometries (Figure 8). Differences were assumed to be statistically significant if the *p*-value was ≤0.05.

An analysis between the cumulative release of RhB and the surface area of the tested devices during a 21-day evaluation period is shown in Figure 8. The RhB release was observed to occur steadily from day 1 to day 21. Specifically, the PCRS sphere exhibited a release of approximately 18% of the total RhB content, ultimately discharging 10.843 μg of the 60 μg of total RhB content in the device over a three-week period. Similarly, the PCRS sphere through-hole showed a release of approximately 22% of the total RhB content, releasing 12.392 μg of 56 μg of total RhB in the device. The PCRS rod exhibited a release of about 14% of the total RhB content, equivalent to 27.129 μg of the 200 μg total RhB content in the device over three weeks. Lastly, the PCRS tube showed a release of 28% of the total RhB content, equivalent to 42.068 μg of the 200 μg total RhB content in the device over the entire experimental period.

### 3.6. Study Limitations

This study had several limitations. As all work was completed in vitro, it is understood that results may differ in vivo. This is due to many contributions, but the differences between PBS and interstitial fluid, subject movement, and local tissue perfusion would all influence release rates. Another limitation was the differential evaporation of the sink solution during sampling. The beakers used were sealed with Parafilm while in the incubator, but there was variance in how long some were uncovered during sampling. To mitigate the risk of variation in manufacturing, devices were visually inspected for quality defects and weighed to ensure they fell within a tolerance of +/−100 μg. Experiments were conducted within controlled time frames for sample extraction, which were within +/−1 h difference for daily sampling and +/−5 min for hourly sampling. A sampling protocol was set to take samples in the same order every day. Sensor drift in the UV–Vis spectrometer may have influenced results, but calibration against blank and positive controls were conducted prior to every set of measurements to ensure minimal impact. Only a mixed ratio of 1:99 RhB to polymer was investigated, and future studies will explore the combination of polymer composites.

In future experiments, the polymeric implant will be introduced into the targeted tumor cavity area of in vivo experiments using a 27-gauge cannula. The objective is to replicate the anatomical structure of the cervix for localized drug delivery. The in vivo experiments will encompass the execution of tests and observations within a living organism to evaluate the efficacy of local drug delivery.

## 4. Conclusions

In this study, an innovative implantable platform made from biodegradable polymers termed the PCRS was developed and characterized. Release of RhB from PCRS devices in vitro was investigated, with PBS serving as a physiological fluid analog. A batch manufacturing process utilizing a metal mold injection technique was developed and experimentally validated to achieve reproducible device geometries with a homogenous mixture of RhB and the polymer base. Cumulative release experiments demonstrated that spherical devices made from PCL demonstrated a consistent biphasic linear release profile over a 21-day period. Further experiments examining different device geometries demonstrated that the mass release rate of RhB was directly proportional to surface area, allowing potential tuning of the release to meet patient-specific needs. The preliminary data obtained in this study indicates that a PCL-based implantable device can achieve a biphasic, loading, and maintenance delivery profile over at least a three-week period. Taken together, results from this study motivate additional study of PCL-based drug delivery systems for minimally invasive placement and localized delivery. This information proves promising to the hypothesis of developing a multistage polymer drug delivery system to exhibit a personalized loading and maintenance phase.

## 5. Patents

Guda, T., Hood, G., Akhter, F., Hood, R.L., Pearson, J., Multistage Polymer Therapeutic Delivery System, US Utility Patent, 16/211,214, 2018.

## Figures and Tables

**Figure 1 pharmaceutics-15-01795-f001:**
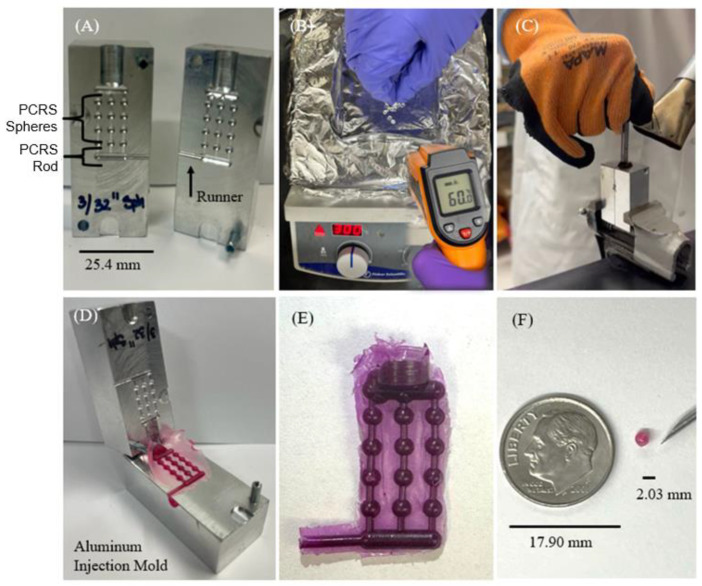
Injection Mold Process (**A**) Aluminum metal mold (**B**) Melting process over an aluminum hot plate (**C**) Polymer injection mold process (**D**) Metal mold opened to show polymer casting (**E**) Polymer raw casting (**F**) Displays the size of a PCRS sphere device in comparison with a US dime.

**Figure 2 pharmaceutics-15-01795-f002:**
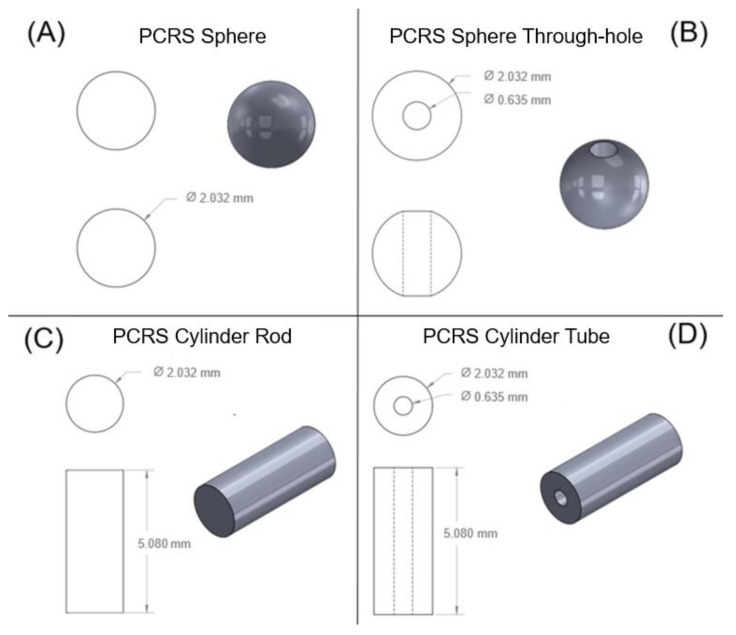
Four different geometrical shapes (**A**) PCRS sphere model with dimensions; (**B**) PCRS sphere through-hole model with dimensions; (**C**) PCRS Cylindrical rod model with dimensions; (**D**) PCRS Cylindrical tube model with dimensions.

**Figure 3 pharmaceutics-15-01795-f003:**
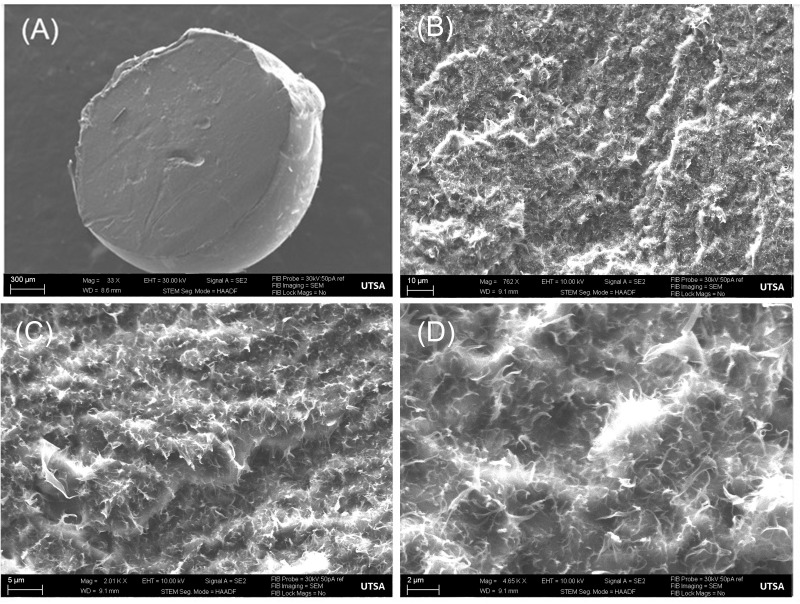
Focused ion beam-scanning electron microscopy images (**A**) A PCRS solid sphere at 300 μm (33×), (**B**) PCRS solid sphere at 10 μm (762×), (**C**) PCRS solid sphere at 5 μm (2010×), (**D**) PCRS solid sphere at 2 μm (4650×).

**Figure 4 pharmaceutics-15-01795-f004:**
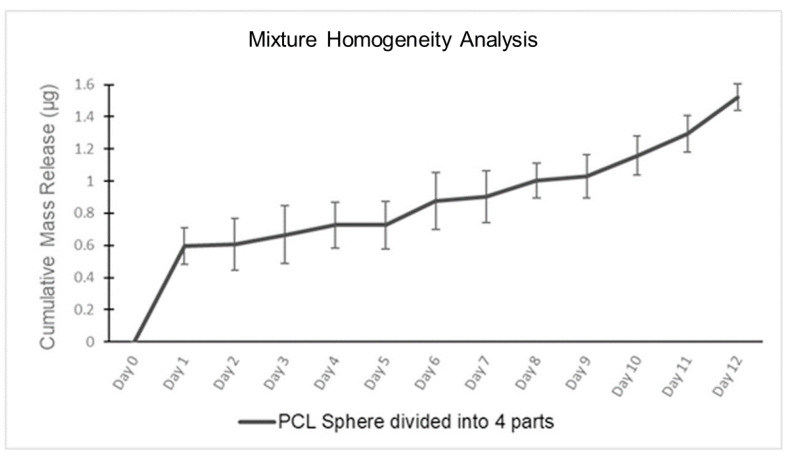
Mixture Homogeneity Analysis: Cumulative RhB release within a split PCRS device.

**Figure 5 pharmaceutics-15-01795-f005:**
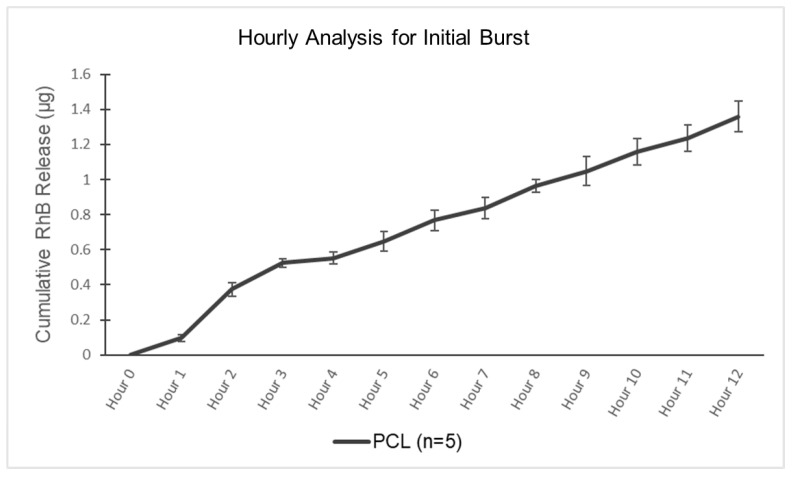
Initial Burst Analysis—Hourly analysis for cumulative RhB release from spherical devices for the initial 12 h following submersion in PBS.

**Figure 6 pharmaceutics-15-01795-f006:**
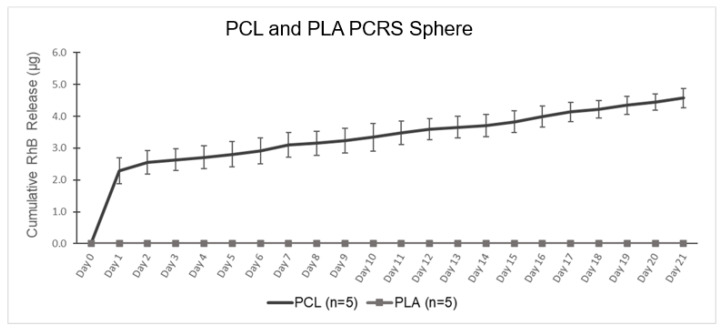
PCL and PLA Single Spherical Device—Graph of the cumulative RhB release of PCL and PLA Single PCRS sphere.

**Figure 7 pharmaceutics-15-01795-f007:**
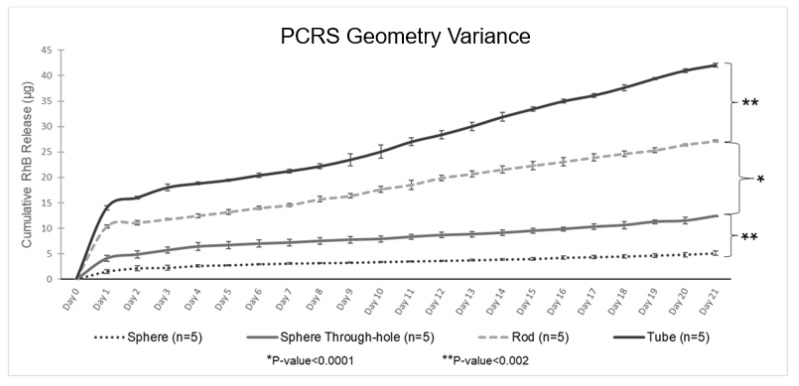
PCRS Geometry Variance—Geometrical device cumulative RhB release analysis for day 21.

**Figure 8 pharmaceutics-15-01795-f008:**
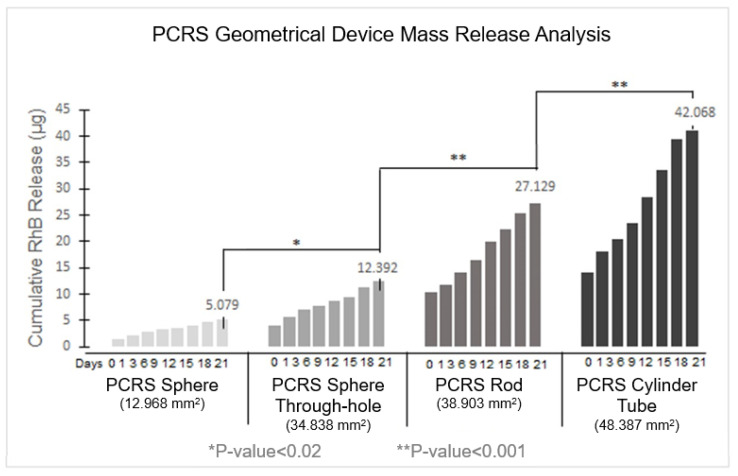
Cumulative Mass Release of RhB: Correlation between the surface area and release for 21 days.

**Table 1 pharmaceutics-15-01795-t001:** Summary dimensions related to the PCRS geometry devices being tested.

Devices	Surface Area (mm^2^)	Dimensions (mm)	Center through- Hole (mm)	Average Mass Weight (µg)	Mass Weight Tolerance (µg)	99% Polymer (µg)	1% RhB (µg)	% Release at 21 Days
PCRS Sphere	12.968	⌀ 2.032	N/A	6000	±100	5940	60	18.07%
PCRS Sphere Through-hole	16.387	⌀ 2.032	⌀ 0.635	5600	±100	5544	56	22.13%
PCRS Cylinder Rod	38.903	5.080 × 2.032	N/A	15,100	±100	14,949	151	13.56%
PCRS Cylinder tube	48.3870	5.080 × 2.032	⌀ 0.635	20,000	±100	19,800	200	27.86%

**Table 2 pharmaceutics-15-01795-t002:** Summary table of the mixture homogeneity analysis.

Phase	Days	Number of Samples	Initial Mass Release (μg)	Final Mass Release (μg)	R^2^ Value
I	0–1	4	0.000	0.596	N/A
II	2–12	4	0.596	1.523	0.932

## Data Availability

Data is available on request to the Corresponding Author.

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
