# Peer review of "An Innovative Polymeric Platform for Controlled and Localized Drug Delivery"

_pharmaceutics, 2023, doi:10.3390/pharmaceutics15071795_

Round 1

Reviewer 1 Report

The main purpose of this work is to design a drug delivery system for the treatment of cervical cancer, but there is nothing about the treatment of cervical cancer in this work. Only a sustained-release system was designed, and no model antitumor drug was used. Therefore, the title and preface of this paper are not appropriate.

In the experiment of Geometry Variance Analysis, please list the detailed process of the experiment. What is the weight of the sample? Which sample has a larger surface area with the same sample weight? In addition, what was the loading content of RhB in each sample

The properties of PCL and PLA are related to molecular weight, please provide the detailed information of both PCL and PLA

Author Response

First, we would like to thank you for the time and effort you spent reviewing our manuscript. We greatly value all of your feedback and believe that our manuscript has been significantly improved by this process. We have addressed all of the Reviewers’ comments; the original comments are numbered below in italic text, and our responses are notated in bullet points. Our revisions to the manuscript are shown in gray-highlighted text in both this document and the manuscript.

Reviewer 2 Report

It is a good paper that will be of interest of the scientific readers in the area of controlled or sustained drug delivery system.  The highlight of geometry of devices in drug delivery is important.  There are no issues in the quality of presentation based on the results obtained in the current study, and therefore the authors are commended for this.  The manuscript is suitable to be published in the Pharmaceutics journal in its current form.

Author Response

First, we would like to thank you for the time and effort you spent reviewing our manuscript. We greatly value all of your feedback and believe that our manuscript has been significantly improved by this process.

Reviewer 3 Report

It is a well written manuscript, covering some interesting research. I´d like to see the at least in vitro toxicity data, but this might be unnecessary as both polymers are biodegradable (one is even approved by the FDA). Or some pilot in vivo results.

However, I recommend the Editors to accept the manuscript as it is.

Author Response

First, we would like to thank you for the time and effort you spent reviewing our manuscript. We greatly value all of your feedback and believe that our manuscript has been significantly improved by this process. We have addressed all of the Reviewer’s comments; the original comments are numbered below in italic text, and our responses are notated in bullet points.

Reviewer 4 Report

1. The abstract and introduction are quite jargon heavy, containing too much broad description. Please make a more clear and detailed one based on the proposed strategy and the results.

2. In the introduction, the authors mainly focus on the drug cisplatin. The description on toxicity of cisplatin is not so accurate. The primary toxicity issue is kidney. The authors also should compare the proposed delivery technology with others (https://doi.org/10.1016/j.jconrel.2022.03.049).

3. What do you mean mulitistage delivery system? It is only a controlled release system.

4. How can this delivery platform be placed in vivo?

5. Some therapeutic results are needed. Only drug release is studied in current version.

Author Response

First, we would like to thank you for the time and effort you spent reviewing our manuscript. We greatly value all of your feedback and believe that our manuscript has been significantly improved by this process. We have addressed all of the Reviewers’ comments; the original comments are numbered below in italic text, and our responses are notated in bullet points. Our revisions to the manuscript are shown in gray-highlighted text in both this document and the manuscript. Thank you for your feedback and for making our manuscript better for readers.

Round 2

Reviewer 4 Report

Although the therapeutic results are not supplied, the authors discussed the future possibility. The authors addressed other concerns well.